# Tenogenic Contribution to Skeletal Muscle Regeneration: The Secretome of Scleraxis Overexpressing Mesenchymal Stem Cells Enhances Myogenic Differentiation In Vitro

**DOI:** 10.3390/ijms21061965

**Published:** 2020-03-13

**Authors:** Maximilian Strenzke, Paolo Alberton, Attila Aszodi, Denitsa Docheva, Elisabeth Haas, Christian Kammerlander, Wolfgang Böcker, Maximilian Michael Saller

**Affiliations:** 1Experimental Surgery and Regenerative Medicine (ExperiMed), Department of General Trauma and Reconstructive Surgery, Ludwig-Maximilians-University (LMU), Fraunhoferstraße 20, 82152 Planegg-Martinsried, Germany; maximilian.strenzke@med.uni-muenchen.de (M.S.); paolo.alberton@med.uni-muenchen.de (P.A.); attila.aszodi@med.uni-muenchen.de (A.A.); elisabeth.haas@med.uni-muenchen.de (E.H.); christian.kammerlander@med.uni-muenchen.de (C.K.); wolgang.boecker@med.uni-muenchen.de (W.B.); 2Experimental Trauma Surgery, Department of Trauma Surgery, University Regensburg Medical Centre, 93053 Regensburg, Germany; denitsa.docheva@ukr.de; 3Division of Hand, Plastic and Aesthetic Surgery, Ludwig-Maximilians-University (LMU), Pettenkoferstraße 8a, 80336 Munich, Germany

**Keywords:** tendon-muscle crosstalk, myotendinous junction, regenerative medicine, skeletal muscle, scleraxis, mesenchymal stem cells, satellite cell, RNA-Seq

## Abstract

Integrity of the musculoskeletal system is essential for the transfer of muscular contraction force to the associated bones. Tendons and skeletal muscles intertwine, but on a cellular level, the myotendinous junctions (MTJs) display a sharp transition zone with a highly specific molecular adaption. The function of MTJs could go beyond a mere structural role and might include homeostasis of this musculoskeletal tissue compound, thus also being involved in skeletal muscle regeneration. Repair processes recapitulate several developmental mechanisms, and as myotendinous interaction does occur already during development, MTJs could likewise contribute to muscle regeneration. Recent studies identified tendon-related, scleraxis-expressing cells that reside in close proximity to the MTJs and the muscle belly. As the muscle-specific function of these scleraxis positive cells is unknown, we compared the influence of two immortalized mesenchymal stem cell (MSC) lines—differing only by the overexpression of scleraxis—on myoblasts morphology, metabolism, migration, fusion, and alignment. Our results revealed a significant increase in myoblast fusion and metabolic activity when exposed to the secretome derived from scleraxis-overexpressing MSCs. However, we found no significant changes in myoblast migration and myofiber alignment. Further analysis of differentially expressed genes between native MSCs and scleraxis-overexpressing MSCs by RNA sequencing unraveled potential candidate genes, i.e., extracellular matrix (ECM) proteins, transmembrane receptors, or proteases that might enhance myoblast fusion. Our results suggest that musculotendinous interaction is essential for the development and healing of skeletal muscles.

## 1. Introduction

Myotendinous junctions (MTJs) are sharp transition zones between tendon and muscle tissue and are ideally suited for the transmission of contractile forces. However, previous developmental studies confirm the assumption that a mere structural role for one another does not seem to explain musculotendinous interaction sufficiently. During somite segregation, myotome-derived fibroblast growth factor 8 (FGF8) already induces the expression of the bHLH transcription factor scleraxis (*SCX)* in adjacently forming syndetomes [1]. Results of a study on avian hind limb revealed that although the early formation of tendon and muscle primordia occur independently, later segregation of individual muscles and tendons depends on their reciprocal influence [2]. Surgical segregation at the MTJ in avian limbs led to a rapid loss of the longitudinal and parallel alignment of myofibers that was accompanied by apoptosis, showing that crosstalk of these cell types occurs already in ontogenesis [3]. As several developmental mechanisms are partly re-enacted during regeneration, musculotendinous crosstalk might also be a component of the muscle healing process [4]. Mesenchymal stem cells (MSCs) are characterized by a rather uncommitted cell stage (within the mesodermal derivate) and might be able to regulate some mechanisms that are shared by myogenesis and regeneration. Previous research revealed various cell types of muscular and non-muscular origin, as well as various biochemical signals that are involved in the three major overlapping phases of muscle regeneration: inflammation, tissue reconstruction, and remodeling [4]. Quiescent satellite cells (SCs), residing beneath the basal lamina along the muscle fibers, are activated by muscle-intrinsic signals after injury and start to regenerate damaged myofibers [4,5,6,7]. However, paracrine signals from non-muscular cells, such as infiltrating regulatory T-cells [8] or macrophages [9], spatiotemporally secret pro- and anti-inflammatory cytokines, including tumor necrosis factor (TNF)-alpha, fibroblast growth factor 2 (FGF2), insulin-like growth factor 1 (IGF1) and hepatocyte growth factor (HGF) that stimulate myoblasts from proliferation to differentiation [9,10,11,12]. In addition, connective tissue cells also play an important role in skeletal muscle regeneration [13]. Fibro/adipogenic progenitors (FAP) reside in quiescence within the interstitial space of skeletal muscle until activation by muscle damage [14]. FAPs then contribute to the development and regeneration process by transient fibrogenesis in close interaction with muscle precursors that prevent excessive extracellular matrix (ECM) production by exosomal miRNA [15,16]. Their activity depends on the expression levels of the angiopoetin-1 receptor (Tie2) and/or vascular cell adhesion molecule 1 (VCAM-1) [17]. Dysregulation of intracellular pathways of FAPs also seems to be involved in excessive fibrotic tissue formation during muscle regeneration and thereby causing impaired healing [18]. A variety of studies also shows the complex interplay of cells and signaling mechanisms that are involved in muscle regeneration. For instance, macrophages and TNFα signaling regulate FAPs appearance and activity in skeletal muscle [17]. A recent detailed study by Giordani et al. revealed different populations of mono-nucleated cells in adult skeletal muscles, including FAPs, integrin alpha 7-positive/vascular cell adhesion molecule 1-negative (ITGA7+/VCAM-1-) cells, lymphocytes, and leukocytes [19]. Interestingly, a previously unknown cell population with a high expression of scleraxis (*Scx*), a gene characterizing tendon (and ligament) progenitor cells from early somite stage on, was reported to reside between myofibers outside the basal lamina [20]. These *Scx* positive cells could potentially contribute to the healing process. However, it remains uncertain if and how these intramuscular *Scx* positive cells contribute to skeletal muscle regeneration.

Therefore, we used *SCX* overexpressing MSCs (Single clone pick 1 (SCP1)*^GFP/SCX^*), which sufficiently mimic the tendon cell lineage, to investigate the tenogenic influence on cellular mechanisms of myoblasts [21]. To unravel the mechanism by which *SCX* influences intracellular downstream pathways in MSCs, and thereby the secretome, we performed a bioinformatics analysis of RNA-Seq data and discuss candidate genes that might cause the observed effects in myoblasts.

## 2. Results

### 2.1. Prolonged Exposure to Conditioned Media from SCP1 Cells Affects the Morphology and Metabolic Activity of Myoblasts

As the secretome of immortalized mesenchymal stem cells that stably overexpress the reporter gene green fluorescent protein (GFP) (SCP1*^GFP^)* or additionally the transcription factor scleraxis (SCP1*^GFP/SCX^*) [21] might differentially influence the morphology and metabolic activity of myoblasts, when compared to normal media (NM), we measured the area, as well as the aspect ratio and quantified the mitochondrial dehydrogenase activity of the myoblasts. While the morphology of the murine myoblastic C2C12 cells did not show any obvious difference after a relatively short incubation time of 4 hours in all media, 20 hours later, myoblasts that were cultured in conditioned medium (CM) derived from SCP1*^GFP^* and SCP1*^GFP/SCX^* displayed slightly smaller appearance and less distinct filopodia (Figure 1A, arrowheads). Quantification of the cell area showed an approximate 20% reduction in size for C2C12 cells in either CM derived from SCP1*^GFP^* or SCP1*^GFP/SCX^* cells, with no significant difference between both CM (Figure 1B). As not only the area is an important descriptor for cell morphology and therewith function, we additionally measured the aspect ratio (Figure 1C). However, none of the culture conditions led to any significant changes in the aspect ratio, independent of the duration of exposure to different CM. As for cell shape, also metabolic activity of myoblasts was not affected by short-term exposure to any of the CM. Instead, long-term exposure to CM of SCP1^GFP^ or SCP1*^GFP/SCX^* cells significantly increased the mitochondrial turnover of tetrazolium salt (Figure 1D). Interestingly, myoblasts in CM derived from SCP1*^GFP/SCX^* showed an even higher, significant 15% increase in metabolic activity, when compared to C2C12 cells in CM of SCP1*^GFP^*.

### 2.2. Conditioned Media from both SCP1 Cell Lines Increased the Migration of C2C12

As not just muscle-intrinsic but also muscle-extrinsic signals might affect the migration of regenerative cells, such as satellite cells or myoblasts, we investigated if CM derived from SCP1*^GFP^* or SCP1*^GFP/SCX^* affects the migration behavior of C2C12 cells. Interestingly, time-lapse analysis of myoblasts that were exposed to CM derived from both SCP1 cell lines (Figure 2A’,A’’) led to an obvious increase in random migration over 24 hours, indicated by the Euclidian distance in a polar plot, when compared to myoblasts in NM (Figure 2A). While the maximum velocity of C2C12 was not affected by CM (Figure 2C), quantification showed a significant 14% increase in the mean velocity, when compared to myoblasts in NM (Figure 2B).

### 2.3. The Secretome of SCP1^GFP/SCX^ Cells Led to An Improved Myoblast Fusion

After muscle injury, quiescent satellite cells under the basal lamina of the myofiber start to proliferate and to differentiate into myoblasts that ultimately fuse with damaged, mature muscle fibers to regenerate the defect [22]. Therefore, we investigated how CM derived from SCP1*^GFP^* and SCP1*^GFP/SCX^* cells affects the fusion of C2C12 cells. After 5 days of myogenic differentiation, myoblasts that were cultured in NM or that were exposed to CM of both SCP1 cell lines formed multi-nucleated myofibers (Figure 3A–A’’, arrowheads). However, C2C12 cells with CM from SCP1*^GFP/SCX^* showed more and longer myotubes (Figure 3A’’), when compared to C2C12 in NM or CM from SCP1*^GFP^* cells (Figure 3A,A’). Remarkably, quantification of the relative myosin heavy chain 1E (MYH1E) positive area by immunofluorescence staining revealed an approximately 47% increase in differentiated myoblasts in CM from SCP1*^GFP/SCX^* cells when compared to myoblasts in NM or CM of SCP1*^GFP^* (Figure 3B). In addition, newly formed myofibers that were treated with CM from SCP1*^GFP/SCX^* cells showed an approximate 2.5-fold increase in nuclei in MYH1E positive cells (Figure 3C).

### 2.4. Newly Formed Myofibers Align to SCP1 Cell Pellets

Not only the fusion itself but also the proper orientation of myofibers is crucial for the overall healing process after muscle injury [23]. As tendon progenitor cells can influence the alignment of forming myofibers during development in vivo [24], we investigated the effect of released SCP1 cytokines on the alignment of fusing myocytes by seeding C2C12 around SCP1*^GFP^* or SCP1*^GFP/SCX^* cell pellets, which should function as directed source alignment cues of myoblasts during fusion.

In accordance with myoblast fusion in CM, myoblasts showed an increased fusion around SCP1*^GFP/SCX^* pellets, when compared to myofibers around SCP1*^GFP^* pellets (Figure 4A). Interestingly, the majority of myofibers were orientated towards the center of the pellets (Figure 4A, dashed circle). Quantification of the angle between the myofibers and the center in a radius of 2000 µm revealed a mean orientation angle of 140.9 ± 28.0°, with a non-significant difference between both experimental groups (Figure 4B).

### 2.5. Scleraxis-Overexpression in SCP1 Cells Leads to Profound Transcriptional Changes

During skeletal muscle repair, regenerative cells are exposed to intrinsic and extrinsic factors that considerably modulate the crosstalk between these cells. Even though many molecular mechanisms that are involved in muscle repair have been identified, the comprehensive complexity of the overall regeneration process is not fully understood. To get the first hints about which factors, derived from surrounding cells of the tendon lineage, may be beneficial for the repair process, we performed deep sequencing of the transcriptome of SCP1*^GFP^* and SCP1*^GFP/SCX^* pellets. Differential gene expression analysis revealed 1329 and 1089 genes that were significantly up- and downregulated, respectively, in SCP1*^GFP/SCX^* pellets when compared to SCP1*^GFP^* pellets (Figure 5A, Appendix A). The 10 most up- and downregulated genes are shown in Table 1.

Hierarchical cluster analysis of the 100 most differentially expressed genes in SCP1*^GFP/SCX^* pellets revealed a clear separation of the two cell lines (Figure 5B). Interestingly, while the analysis of the second cluster did not show any overrepresented biological function analyzed by gene ontology, 18 out of 60 significant upregulated genes in the first cluster are involved in the immune response. Furthermore, the evaluation with gene ontology of all significantly upregulated genes in SCP1*^GFP/SCX^* cells revealed a plentitude of different biological functions, such as cell communication, signaling, and locomotion (Figure 5C). The whole list of changed gene ontologies is provided in Appendix A.

To specify which potential pathways are involved in the cellular response of C2C12 cells to CM derived from SCP1*^GFP/SCX^* cells in comparison to CM of SCP1*^GFP^* cells, we performed a STRING analysis [25] and manually investigated interaction clusters by literature search (Figure 5D, labels). Interestingly, beside cell-intrinsic pathway regulation, such as suppression of cytokine or Rho-dependent signaling, several potential pathways relevant for regeneration, such as collagen turnover or regulation of cell migration, were changed in SCP1*^GFP/SCX^* cells, when compared to SCP1*^GFP^* cells.

## 3. Discussion

Many aspects of the musculotendinous crosstalk during development have been discovered so far [1,2]. These insights lead to the assumption that the function of muscle-tendon compounds go beyond a mere structural role, and as muscle and tendon strongly interact during ontogenesis, these developmental mechanisms might also be involved in muscle homeostasis and regeneration. We believe that one of the key non-myogenic cellular contributors could be *Scx*-positive cells that were found within the interstitial space of skeletal muscle fibers [19].

As myofiber formation relies on the cytoskeletal rearrangement of myoblasts [26], the reduction in C2C12 size in response to CM exposure might be part of an initial step of the fusion cascade. Small, round cells are thought to be a distinct subpopulation within primary satellite cultures [6], which might imply a similar cellular adaptation of C2C12 towards a more primitive stem cell state after the exposure to CM from both SCP1 cell lines. Furthermore, in vitro cultured porcine satellite cells gradually increase their size but not their aspect ratio [27]. The increase in metabolic activity, especially after treatment with CM of SCP1*^GFP/SCX^* cells, might be based on a similar mitochondrial adaptation of satellite cells in response to injury, as the activation of these stem cells leads to a switch from a highly glycolytic to oxidative state [28].

Migration of myoblasts and satellite cells is essential during the regeneration of injured muscles, especially in the initial phase of the healing process [6]. While several myoblast-intrinsic cytokines that are connected to migration have already been identified [29], muscle-extrinsic ligands are not fully described. Interestingly, the increase in C2C12 migration in CM of both SCP1 cell lines might also be a result of the transition to a more primitive cell state, as myoblasts reduce their migration during myogenic differentiation [30]. Additionally, a high migration potential of satellite cells is crucial for a non-locally-restricted regeneration of damaged myofibers [31]. Extrinsic proteases, such as MMPs [32,33] or urokinase plasminogen activator [34], positively affect C2C12 migration, especially and it is reasonably certain that various MMPs are released by SCP1 cells. Indeed, our RNA-Seq data showed various highly expressed MMPs, such as *MMP1*, *MMP2,* and *MMP13*.

The ultimate goal of activated satellite cells and myoblasts is the fusion with damaged myofibers and/or to fuse to new myofibers that replace severely injured fibers. Our data revealed a significant increase in myocyte fusion in CM derived from SCP1*^GFP/SCX^*. Our RNA sequencing data revealed that SCP1*^GFP/SCX^*, which mimic the tendon stem/progenitor cells [21], released cytokines and ECM molecules that positively influence myogenic metabolism and fusion of C2C12 cells. Nevertheless, none of the 10 most up- and downregulated genes had a direct paracrine effect on myogenic progression. However, laminin and collagen IV are the major ECM molecules of the basal lamina of muscle fibers [35] and play an important role in regeneration [36], and interestingly, SCP1*^GFP/SCX^* cells showed upregulation of different collagen type IV alpha chains (*COL4A1-4, COL4A2*), as well as laminin subunits (*LAMA5, LAMB2, LAMB3*), when compared to normal SCP1 cells. They might be possible candidates to explain the increased fusion of C2C12 cells in CM derived from SCP1*^GFP/SCX^* cells.

During normal physiological regeneration, exercise improves the ingrowth of new capillaries into the defect side, the regeneration of myofibers, and, interestingly, also leads to more parallel orientated myofibers [37]. However, our results did not reach a significant effect on the alignment of myofibers in vitro.

Various non-myogenic cell types and signaling pathways that contribute and enhance muscle repair are known and thus, can provide the basis for the underlying mechanisms that are involved in the regeneration of skeletal muscles [9,10,14]. Therefore, future studies should focus on the contribution of *Scx* positive tenogenic cells to this complex process. As not only myofiber intrinsic signaling pathways, including cytokines, ECM and ECM modulating molecules, are involved in the activation of satellite cells after injury [38,39], other cells, such as FAPs, neutrophils and macrophages, contribute muscle-extrinsic signals that influence the regenerative potential of myoblasts [40].

The great number of candidate genes and unknown mechanisms for a possible contribution of tendon cells to skeletal muscle regeneration visualize the importance of investigating involved processes in more detail. Future loss- and gain-of-function experiments in vitro and in vivo that compare scleraxis expressing cells’ activity and gene expression patterns within healthy and regenerating skeletal muscle could support our model of tendinous contribution to skeletal muscle repair and reveal more details about the underlying mechanisms.

## 4. Materials and Methods 

### 4.1. Cell Lines, Preparation, and Protein Quantification of Conditioned Medium

For all experiments, the immortalized myoblast cell line C2C12 (Sigma–Aldrich, St. Louis, MO, USA) and the immortalized bone marrow-derived human mesenchymal stem cell lines SCP1*^GFP^* and SCP1*^GFP/SCX^* were used [21,41]. SCP1*^GFP/SCX^* cells ectopically express the tendon-specific transcription factor SCX, thus mimicking tendon progenitor cells [21]. All cell lines were cultured and expanded in normal medium (NM), consisting of DMEM with GlutaMAX^TM^-I (Thermo Fisher, Waltham, MA, USA), 10% heat-inactivated fetal bovine serum (FBS, Sigma–Aldrich, St. Louis, MO, USA), and 40 IU/ml penicillin/streptomycin (Gibco-Thermo Fisher, Waltham, MA, USA) in a humidified incubator at 5% CO2 and 37 °C. The medium was changed at least twice per week if not stated otherwise, and a cell confluence was not allowed to reach over 50% to avoid spontaneous fusion of C2C12 cells. To prepare the conditioned medium (CM), SCP1*^GFP^* and SCP1*^GFP/SCX^* were cultured on T175 cell culture flasks in 20 ml NM, which was renewed every two days until a confluent monolayer was formed (Nunc^TM^, Thermo Fisher, Waltham, MA, USA). Then fresh NM was added and withdrawn after 24 hours, centrifuged for 10 minutes at 1000× *g,* and the supernatant was mixed in a 1:1 ratio with fresh NM and stored in aliquots at −20 °C until use. For protein content quantification, CM was produced equivalently except for using phenol red free DMEM (Thermo Fisher, Waltham, MA, USA) without FBS, enriched with 1% GlutaMax^TM^-I, 1% pyruvate, and 40 IU/mL penicillin/streptomycin. After 24 h incubation, the media were collected, centrifuged for 5 minutes at 500 RPM, and the supernatant was stored until use. To validate equal protein concentrations of CM, we quantified protein content using spectral photometry using a NanoDrop^TM^ Lite Spectral Photometer (Thermo Fisher, Waltham, MA, USA).

### 4.2. Evaluation of Cell Morphology and Metabolic Activity

For the quantification of morphological changes, C2C12 at a density of 1000 cells/cm^2^ were incubated with CM of SCP1*^GFP^* or SCP1*^GFP/SCX^,* and images were taken in five different field-of-views (FOV) per well, at 4 and 24 hours after CM exposure. The area and aspect ratio (major axis/minor axis) of all cells within each FOV were quantified with Fiji [42]. For metabolic activity, 5000 C2C12 were seeded in a 24-well plate and incubated with CM mixed in 1:10 ratio with the Single clone pick 1 (WST-1) solution (Roche, Penzberg, Germany). After 4 hours, the colorimetric conversion of the WST-1 was quantified with a Multiskan^TM^ FC (Thermo Fisher Scientific, Waltham, MA, USA) at 450 nm. All values were normalized in each timeframe to myoblasts in NM.

### 4.3. Migration Assay

To evaluate potential changes on the random migratory behavior of C2C12 in response to CM of SCP1*^GFP^* or SCP1*^GFP/SCX^*, myoblasts were seed at a density of 1000 cells/cm^2^ with NM and medium was changed after 3 hours to CM. Images were acquired every 15 minutes over 24 hours in six random FOVs per well. All cells that did not leave the FOV were manually tracked, and the mean, as well as the maximum velocity, was calculated with the open-source plugin MTrackJ [43] for ImageJ.

### 4.4. Myoblast Fusion Assay

C2C12 were seeded at a density of 5000 cells/cm^2^ and incubated for 24 hours in NM. To initialize myogenic differentiation, cells were washed once with PBS, and fresh CM (as described in paragraph 4.1) was added without any additional supplements. After 3 days, CM medium was exchanged with fresh CM. Five days after the first CM exposure, newly formed myotubes were visualized by immunohistochemistry against myosin heavy chain 1E (MYH1E, clone MF20, obtained from the Developmental Studies Hybridoma Bank (DSHB) developed under the auspices of the National Institute of Child Health and Human Development (NICHD) and maintained by The University of Iowa, Department of Biological Sciences, Iowa City, IA 52242). Therefore, cells were fixed for 15 minutes in 4% paraformaldehyde (PFA) in PBS, washed three times in PBS for 5 minutes and blocked for 1 hour in blocking solution (0.5% Triton X100 and 10% HS in PBS) all at room temperature. Afterward, cells were incubated with the primary antibody (1:50 in blocking solution) overnight at 4 °C, washed three times with PBS, and incubated with a fluorophore-conjugated secondary antibody (1:250 in blocking solution, Thermo Fisher, Waltham, MA, USA) for 1 hour at room temperature. Finally, cells were washed two times in PBS, counterstained with 4′,6-Diamidine-2′-phenylindole dihydrochloride (DAPI), washed once more with PBS and mounted in Fluoroshield (Abcam, Cambridge, MA, USA). Large overview images (approx. 12 mm^2^) were semi-automatically acquired with an epifluorescence microscope (AxioObserver, Carl Zeiss, Jena, Germany). While the relative area of fused myoblasts was quantified with ImageJ by applying the same threshold in all images, the fusion index was obtained by dividing the number of all nuclei in the whole image by the number of nuclei within MYH1E positive cells with three or more nuclei.

### 4.5. Myofiber Alignment Assay

SCP1*^GFP^* and SCP1*^GFP/SCX^* cells were pelleted by centrifugation of 50,000 cells in a non-cell culture treated v-bottom 96-well plate (Corning, Amsterdam, The Netherlands) at 400 g for 10 minutes. After 24 hours in NM, compacted pellets were randomly distributed in a T25 flask, and spheroids were allowed to adhere in a reduced amount of medium for 3 hours. Afterward, C2C12 cells were added at a density of 10,000 cells/cm^2^ in differentiation medium, consisting of DMEM with GlutaMAX^TM^-I, 2% heat-inactivated horse serum (HyClone, GE Healthcare, Chicago, IL, USA) and 40 IU/mL penicillin/streptomycin. After 5 days of differentiation, cells were fixed, stained against MYH1E, and imaged as described above. Fresh medium was added once after 3 days. Alignment of newly formed MYH1E positive myofibers was evaluated with ImageJ, by measuring the angle between the longitudinal axis and the center of the pellet of all myofibers within a 2 mm range around the cell pellets. Thus, 180° corresponds to a myofiber that is perfectly aligned towards the center of the pellet.

### 4.6. RNA-Sequencing and Bioinformatics

Cell pellets were created as described above, cultured for an additional 24 hours, and directly lysed in Trizol (Invitrogen, USA). RNA was isolated from three independent preparations following a standard protocol. RNA quality was measured with a BioAnalyzer (Agilent Technologies, Santa Clara, CA USA), and libraries for sequencing were prepared with a SENSE mRNA-Seq Library Prep Kit V2 (Lexogen, Vienna, Austria). All libraries were sequenced on a HiSeq1500 device (Illumina, San Diego, CA, USA) with a read length of 50 bp and a sequencing depth of approximately 20 million reads. After demultiplexing, reads were aligned to the human reference genome (version GRCH38.92) with STAR [44]. Differential gene expression analysis was performed using DESeq2 [45], and an adjusted *p*-value of less than 0.05 was set to determine significant changed genes between SCP1*^GFP^* and SCP1*^GFP/SCX^* cell pellets. Gene ontology and functional protein association networks were visualized with REVIGO or STRING, respectively [25,46]. Furthermore, RNA-Seq data were examined to spot possible candidate genes.

### 4.7. Statistical Analysis

All experiments were carried out at least three times in triplicates. Statistical significance was calculated after the determination of a Gaussian distribution using either a one-way ANOVA-test or a *t*-test with appropriate post-hoc tests (GraphPad Prism, San Diego, CA, USA). Statistical significance was assumed at a *p*-value of ≤ 0.05. Data are represented as either the mean and the standard deviation (SD) or the median with quartiles.

## 5. Conclusions

Taken together, the secretome of scleraxis-overexpressing MSCs (SCP1*^GFP/SCX^*) positively affects the metabolic activity and fusion capacity of myoblasts by a yet unknown mechanism. It implies a supportive role of interfibrillary residing cells of the tendon lineage via scleraxis downstream targets for skeletal muscle repair. Future loss and gain-of-function experiments of promising candidate genes, which we determined by RNA sequencing, might reveal specific targets, that have a positive effect on the local microenvironment in the defect site during skeletal muscle regeneration. We consider our findings to be useful for further approaches to the knowledge of muscle repair, which may be beneficial for progress in therapy development for muscular diseases and injuries.

## Figures and Tables

**Figure 1 ijms-21-01965-f001:**
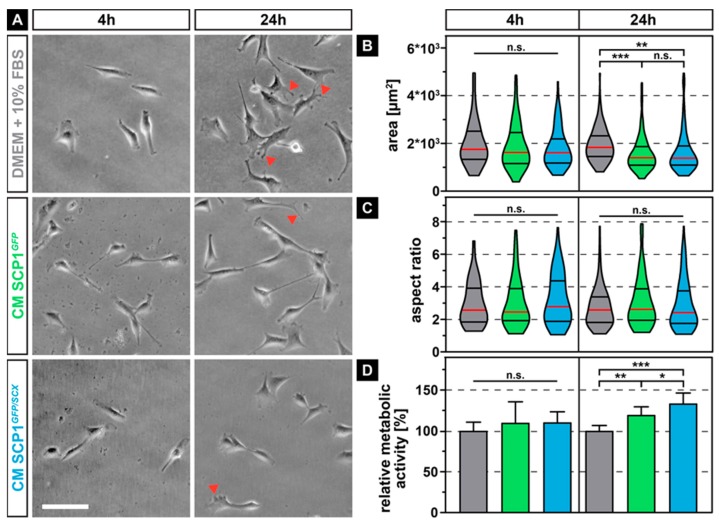
Phase contrast images of C2C12 cells after 4 and 24 hours of exposure to conditioned media (CM) derived from Single clone pick 1 (SCP1)*^GFP^* or SCP1*^GFP/SCX^* cells led to smaller cells with fewer filopodia (arrowheads) when compared to myoblasts in normal medium (NM) (**A**). Quantification of the individual cell morphology parameters revealed a significant decrease in cell area (**B**) but not of the aspect ratio (**C**). Long-term treatment (24 h) significantly increased the metabolic activity of C2C12 in CM of SCP1*^GFP^* or SCP1*^GFP/SCX^* cells, when compared to myoblasts in normal medium. Interestingly, the secretome of scleraxis-overexpressing cells led to a higher metabolic activity of C2C12 cells, when compared to CM from SCP1*^GFP^* (**D**). Scale bar: 100 µm. Violin plots represent the median (red line) and quartiles (black lines). Bar plots represent the mean and SD. * equals *p* ≤ 0.05, ** equals *p* ≤ 0.01, *** equals *p* ≤ 0.001, n.s.: not significant. Data were obtained from three randomly selected field-of-views (FOV) of three independent experiments. At least 75 cells per time point and experimental condition were evaluated.

**Figure 2 ijms-21-01965-f002:**
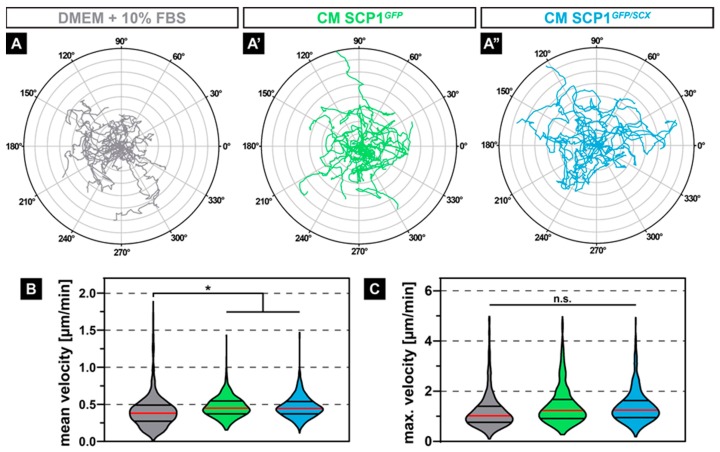
Polar plots of 20 exemplary cell migration tracks showed increased random migration and a longer Euclidian distance of myoblasts treated with conditioned media (CM) derived from SCP1*^GFP^* or SCP1*^GFP/SCX^* cells when compared to C2C12 in standard culture media (**A–A’’**). Quantification of the migratory parameters confirmed the subjective appearance of the cell tracks, by revealing a significant increase in the mean velocity (**B**) with an unchanged maximum velocity (**C**). Distance of concentric circles: 50 µm. Vertical lines in violin plots represent the median (red) and quartiles (black). * equals *p* ≤ 0.05, n.s.: not significant. Data were obtained from three randomly selected field-of-views (FOV) of three independent experiments. At least 530 cells per experimental condition were evaluated.

**Figure 3 ijms-21-01965-f003:**
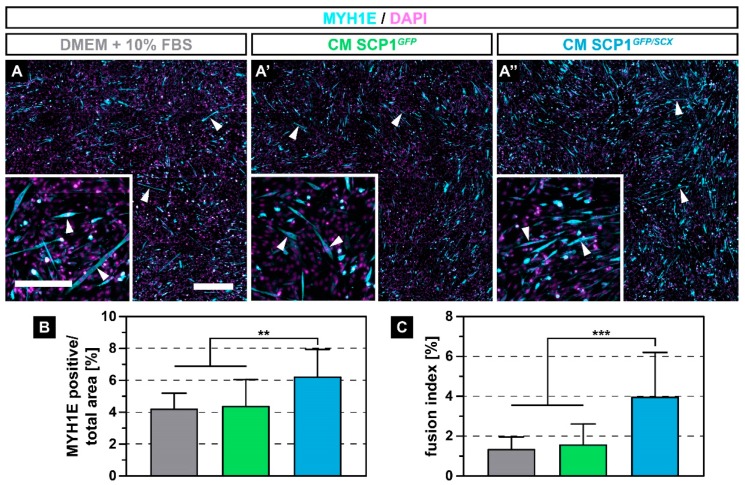
Immunohistochemistry against myosin heavy chain 1E (MYH1E) (turquoise) after 5 days of myogenic differentiation showed newly fused myotubes in all culture conditions (**A–A’’**, arrowheads). However, myoblasts that were exposed to conditioned medium (CM) derived from SCP1*^GFP/SCX^* cells showed an increased fusion, when compared to C2C12 in NM or CM of SCP1*^GFP^* cells (A-A’’). Quantification of the MYH1E positive area (**B**) and the fusion index (**C**) validated the microscopic appearance. Scale bars: 500 µm (overview), 200 µm (insert). Bar plots represent the mean and SD. ** equals *p* ≤ 0.01, *** equals *p* ≤ 0.001. Five randomly selected pictures of three independent experiments were analyzed.

**Figure 4 ijms-21-01965-f004:**
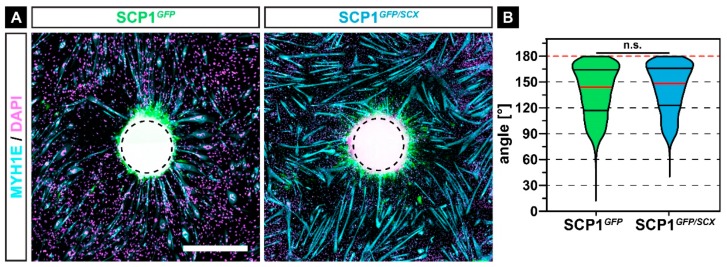
Immunohistochemistry against MYH1E (turquoise) after 7 days of myogenic differentiation in co-culture with SCP1*^GFP^* or SCP1*^GFP/SCX^* cell pellets revealed that most of the newly formed myofibers were orientated towards the middle of the cell pellets (**A**). However, the quantification of the orientation did not show any significant difference between both groups (**B**). Vertical lines in violin plots represent the median (red) and quartiles (black). Red dashed lines indicate a perfect orientation towards the center of the pellet. n.s.: not significant. Scale bar: 500 µm.

**Figure 5 ijms-21-01965-f005:**
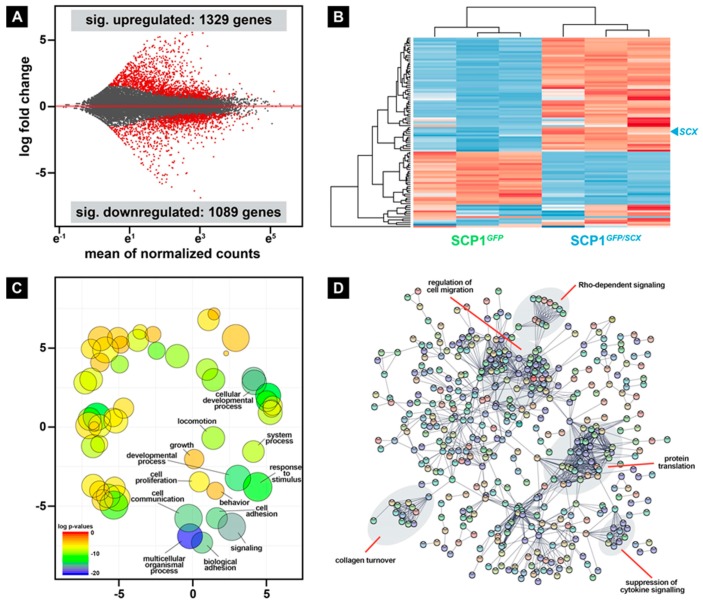
Differential gene expression analysis revealed hundreds of significant up- and downregulated genes in SCP1*^GFP/SCX^* when compared to normal SCP1 (**A**). Furthermore, hierarchical cluster analysis of the 100 most differentially expressed genes clearly separates both cell lines (**B**). Gene ontology analysis of all significantly overexpressed genes in SCP1*^GFP/SCX^* cells revealed several changes in biological functions (**C**) that also include collagen modifying genes, as well as released cytokines that are connected to cell migration (**D**). RNA-Seq data were obtained from three independent pellet preparations.

**Table 1 ijms-21-01965-t001:** The 10 most up- and downregulated genes in SCP1^GFP/SCX^ vs. SCP1^GFP^ cell pellets.

	Gene Symbol	Gene Name	Log2 (Fold Change)
10 most upregulated genes	KCNC2	Potassium Voltage-Gated Channel Subfamily C Member 2	10.81
	PLXDC2	Plexin Domain Containing 2	10.49
	SLC4A3	Solute Carrier Family 4 Member 3	8.52
	AC097478.1	Novel Transcript, Antisense To SNCA	8.31
	TSHZ2	Teashirt Zinc Finger Homeobox 2	8.25
	LINC01285	Long Intergenic Non-Protein Coding RNA 1285	8.11
	SCX	Scleraxis	7.94
	AC010615.2	Novel Transcript, long non-coding RNA	7.69
	RNF212	Ring Finger Protein 212	7.65
	PON1	Paraoxonase 1	7.47
10 most downregulated genes	DACH2	Dachshund Family Transcription Factor 2	−9.50
	VAT1L	Vesicle Amine Transport 1 Like	−9.46
	KCNJ2	Potassium Inwardly Rectifying Channel Subfamily J Member 2	−9.30
	KYNU	Kynureninase	−9.17
	VCAM1	Vascular Cell Adhesion Molecule 1	−9.05
	ZNF804A	Zinc Finger Protein 804A	−8.77
	SOX5	SRY-Box Transcription Factor 5	−8.39
	CES1	Carboxylesterase 1	−8.29
	MYOCD	Myocardin	−8.22
	AC023301.1	Novel Transcript, Sense Intronic To NETO1	−7.97

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
