# Peer review of "Tenogenic Contribution to Skeletal Muscle Regeneration: The Secretome of Scleraxis Overexpressing Mesenchymal Stem Cells Enhances Myogenic Differentiation In Vitro"

_ijms, 2020, doi:10.3390/ijms21061965_

Round 1
Reviewer 1 Report
The manuscript titled “Tenogenic contribution to skeletal muscle regeneration: The secretome of scleraxis overexpressing mesenchymal stem cells enhances myogenic differentiation” investigates the contribution of myotendinous junctions (MTJs) to muscle regeneration by taking into consideration the presence of tendon-related, scleraxis-expressing cells that were found in close proximity to the MTJs and the muscle belly. In order to clarify the muscle-specific function of these scleraxis (SCX)-positive cells, the Authors compared the influence of two mesenchymal stem cell (MSC) lines - one of which was characterized by the overexpression of SCX - on myoblasts morphology, metabolism, migration, fusion and alignment. The results collected by the Authors confirmed that the exposure to the secretome of SCX-overexpressing MSCs significantly enhaced myoblast fusion and metabolic activity. This suggested the essential role of musculotendinous interaction for the development and
regeneration of skeletal muscles.
The work is quite interesting and falls within the scopus of International Journal of Molecular Sciences. The Authors make clear the intended practical application of the research, as well as its novelty. Manuscript is well written and english language is appropriate. The experimental plan is properly designed and well developed; methods are clearly described and sufficiently detailed. The conclusions are well supported by the results.
Below are some minor issues that still need to be addressed:
- The nomenclature for cell lines (SCP1GFP and SCP1GFP/SCX) should be better explained in the Results section.
- Page 9, paragraph 4.2: how did the author choose the incubation periods of 4 and 24 hours? Shoul be useful to also consider longer incubation periods?
- Page 9, paragraph 4.3: for migratory assay, the Authors indicate that images of cells were acquired every 15 minutes, but no details on total acquisition time are given.
- Page 9, paragraph 4.4: for myoblast fusion assay, were C2C12 cells subjected to 5 days of differentiation treatment (as we understand by the result description in paragraph 2.3, line 143, page 4)? The Authors should better clarify this point, also clearly describing what the differentiation treatment consisted of.
- In this study, the effect of human MSC secretomes is evaluated on murine myoblasts. To better mimic what phisiologically could happen, myoblastic cell line and mesenchymal stem cell lines of the same species should be considered.
Reviewer 2 Report
Dear authors:
It has been a pleasure to review your paper about “Tenogenic contribution to skeletal muscle regeneration: The secretome of scleraxis overexpressing mesenchymal stem cells enhances myogenic differentiation” I think that it is the first time that I have only two suggestions to change , to the rest it has been great to read the paper and the idea is very innovate
- Can you include the type of study in the title?
- Can you include the name of the software used in the statistical analysis?
